# Small-Area Radiofrequency-Energy-Harvesting Integrated Circuits for Powering Wireless Sensor Networks

**DOI:** 10.3390/s19081754

**Published:** 2019-04-12

**Authors:** Guo-Ming Sung, Chao-Kong Chung, Yu-Jen Lai, Jin-Yu Syu

**Affiliations:** 1Department of Electrical Engineering, National Taipei University of Technology, Taipei 10608, Taiwan; james591126@gmail.com (C.-K.C.); boredcity48@gmail.com (Y.-J.L.); j887110@gmail.com (J.-Y.S.); 2Research and Development Centre for Smart Textile Technology, National Taipei University of Technology, 1, Sec. 3, Zhongxiao E. Rd., Taipei 10608, Taiwan; 3General Manager Office, Goodfield Technology Co., Ltd., 46, Lane 55, Sec. 3, Zhiyuan Rd., Taipei 11265, Taiwan

**Keywords:** radiofrequency, ISM 915 MHz, energy-harvesting IC, native MOS, Dickson voltage multiplier, over-voltage protection circuit, low-dropout regulator, charger control circuit

## Abstract

This study presents a radiofrequency (RF)-energy-harvesting integrated circuit (IC) for powering wireless sensor networks with a wireless transmitter with an industrial, scientific, and medical (ISM) of 915 MHz. The proposed IC comprises an RF-direct current (DC) rectifier, an over-voltage protection circuit, a low-power low-dropout (LDO) voltage regulator, and a charger control circuit. In the RF-DC rectifier circuit, a six-stage Dickson voltage multiplier circuit is used to improve the received RF signal to a DC voltage by using native MOS with a small threshold voltage. The over-voltage protection circuit is used to prevent a high-voltage breakdown phenomenon from the RF front-end circuit, particularly for near-field communication. A low-power LDO regulator is designed to provide stable voltage by using zero frequency compensation and a voltage-trimming feedback. Charging current is amplified N times by using a current mirror to rapidly and stably charge a battery in the proposed charger control circuit. The obtained results revealed that the maximum power conversion efficiency of the proposed RF-energy-harvesting IC was 40.56% at an input power of −6 dBm, an output voltage of 1.5 V, and a load of 30 kΩ. A chip area of the RF-energy-harvesting IC was 0.58 × 0.49 mm^2^, including input/output pads, and power consumption was 42 μW.

## 1. Introduction

Energy harvesting for supplying power to low-power electronic devices has recently become mainstream research. The lifetime of a battery can be extended using a developed self-sustainable power supply. Bito et al. designed a flexible wearable radiofrequency (RF) energy harvester for off-the-shelf two-way talk radios of 2 W using inkjet printing technology, and E- and H-field energy harvesters were verified using a light-emitting diode and microcontroller communication module [1]. Tian et al. presents an integrated solution for a flexible direct current (DC)–DC converter by embedding a flexible polyimide printed circuit board and an inductor made of flexible ferrite-polymer composite in a wire [2]. A wearable RF-energy-harvesting device, which comprises a U-shaped dipole antenna, matching network, RF-DC converter, and DC–DC converter, was presented in [3] for supplying power to smart jewelry. This design converts a 915-MHz RF signal into a constant DC output voltage of 3.1 V at an input power of −6 dBm, which is suitable for supplying power to a fitness monitor pendant under standby mode. An experimental comparison in [3] shows that the Dickson topology has high efficiency at a high input power.

A commercial power supply or battery can be used as a stable power source for providing power to wireless sensor networks (WSNs). However, these power supplies have a finite lifetime, have a large size, incur high maintenance cost, and cause environmental pollution. RF energy harvesting was introduced because of potentially long lifespan, low-power consumption, and small size [4]. In this process, ambient energy is emitted from various sources, such as television (TV), radio, wireless Internet, satellite communication, and base stations, which is converted into feasible DC voltages. WSN devices powered by RF energy can be used in several applications, such as telemetry systems, RF tags, home automation, equipment monitoring, access control, and prolongation of lifespan of powered devices [5,6]. In [7], a five-stage orthogonally switching charge pump rectifier with a voltage boosting network was used for an RF energy harvester. The rectifier provides a DC voltage and power conversion efficiency (*PCE*) of 1.3 V and 33.72%, respectively, with a load of 100 kΩ from an RF source of −13 dBm at an ISM frequency.

The WSNs are fabricated using low-power circuits, particularly by integrating with an antenna and a voltage multiplier for a stable power supply [8,9]. A combination of a two-stage Villard multiplier with a three-stage Dickson voltage multiplier was proposed to increase output voltage and power efficiency [10]. A Villard–Dickson power harvester circuit can produce an output voltage of 4 V with an efficiency of 3.9% and an output voltage of 8.8 V with an efficiency of 14.6% without and with a matching circuit, respectively, at an input signal of 4.1 dBm from a TV station [10]. A microstrip matching circuit and zero-biased diode were used to improve the performance of the designed energy-harvesting device.

The Internet of Things (IoT) is a well-known platform, wherein each physical object is connected to the Internet without requiring human interaction. Few people check for regular supply of power to IoT, excluding WSNs. To supply power to these communication devices for a long duration, a self-sustainable power system is required for complex networks [11]. A feasible technique is to harvest RF energy from external environmental sources, which could be used as an auxiliary power supply for increasing the battery life [12,13]. In general, RF energy-harvesting bands from ISM (900–928 MHz) and WLAN (2.4/5 GHz) have received considerable attention because they can cover cities and countries due to their continuous emission as commercial bands [14,15,16]. In [17], RF energy harvesting employed a Wi-Fi band at a frequency of 2.4 GHz by using the P21XXCSR-EVB evaluation board from Powercast. Four types of antennas and an RF-energy-harvesting evaluation board are included in this research. Experiments have shown that power losses in air and antennas have a stronger effect on the total efficiency of the system than the losses in the RF energy harvester. The RF-energy-harvesting system can be improved using antenna arrays, particularly for supplying power to low-power sensors and IoT devices.

Most previous studies have used an individual energy-harvesting source to supply power to low-power sensors and IoT devices. However, the available environmental energies affect the input power of the RF transmitter (Powercast Corporation) at a frequency of 2.4 GHz. Sensors and IoT devices are always installed on the ceiling of a room. For uniform RF transmission and increasing communication distance, this study employed a TX91501 RF transmitter from Powercast Corporation [18], which provides an output power of 3 W at an ISM band of 915 MHz, with a maximum transmission distance of 12–15 m. Moreover, a 1-dBi bipolar antenna was used as a receiving antenna to design an RF-energy-harvesting chip. Figure 1 shows the proposed RF-energy-harvesting system, which includes a Powercast transmitter, receiving antenna, RF-energy-harvesting IC, and power source.

The following section elucidates the circuit topology of the designed RF-energy-harvesting IC. Section 3 and Section 4 present the simulated and measured results and conclusions, respectively.

## 2. Circuit Topology of RF-Energy-Harvesting Integrated Circuit

Figure 2 shows a functional diagram of the RF-energy-harvesting IC, which receives RF energy from the receiving antenna, and this energy charges energy storage devices. Thus, the proposed IC is used as a backup stable power source. An off-chip matching network from MuRata Manufacturing Co. Ltd. was used to guarantee maximum power transfer. The RF-DC rectifier received the transferred RF energy from a matching network and converted this energy into DC voltage. An over-voltage protection circuit was then designed to protect the RF-DC rectifier when the output voltage of the designed rectifier was higher than the breakdown voltage [19]. A stable voltage of 1.5 V is passed through the low-power LDO circuit and sent to the charging control circuit for charging storage devices such as a battery.

### 2.1. Matching Network

Because of an antenna impedance of 50 Ω, the input impedance of the matching network must be matched to 50 Ω. This matching circuit is used to not only guarantee maximum power transfer but also obtain maximum conversion efficiency. If the matching network includes capacitor C and inductor L, maximum power transformation is obtained from an antenna to a load. Source impedance *R_a_*must be equal to the conjugate input impedance *Z_in_* of the matching network [20]. The maximum output power *P_L,max_* can be expressed as follows:(1)$$P_{L,\max}=\left|I_{a}^{2}\right|\times ℜe[Z_{in}]=\frac{V_{a}^{2}}{4R_{a}}$$
where *V_a_* is the peak voltage of the antenna. *I_a_* and *Z_in_* are the input current and input impedance of the matching network, respectively.

In this study, the impedance of the antenna and the input impedance of the RF-DC rectifier were 50 Ω and 240.4 – j473.9 Ω, respectively. The impedance of the matching network must be equal to 50 Ω [21]. Simulations were performed with ADS software and component datasheet provided from MuRata Company, and inductor L, labeled as LQG15HS_02, and capacitor C, labeled as GRM15, were used to complete the matching network. Moreover, authors paid more attention toward self-resonant frequencies (SRFs) for selecting a suitable inductor. The operating frequency of the matching network must be lower than the SRF of the selected inductor. The larger the inductor is, the smaller the SRF is. For example, the minimum SRFs were 1000 and 800 MHz for L ≤ 47 μH and L ≥ 56 μH, respectively. If input power was set to −10 dBm at an operating frequency of 915 MHz, a suitable inductor lower than 47 μH was selected.

### 2.2. RF-DC Rectifier

For an N-stage rectifier, a pair of a metal-oxide-semiconductor field-effect transistor [MOSFET (M_P_)] and capacitor (C_P_) can be considered as a rectifier with a small ripple voltage across C_p_. An averaged output voltage of the (*p* + 1)^th^ stage can be expressed as follows:(2)$$V_{O\left(p+1\right)}\left(V\right)=V_{O\left(p\right)}\left(V\right)+V_{boost}\left(V\right),$$
where *V*_*O*(*p*)_, *V*_*O*(*p*+1),_ and *V_boost_* are the output voltage of the *p*^th^ stage (present voltage), output voltage of the (*p* + 1)^th^ stage (next voltage), and incremental voltage of each stage [22], respectively. Approximation in charge computation was used to provide the incremental voltage *V_boost_* as follows:(3)$$V_{boost}\left(V\right)=V_{in}^{\prime }-V_{tn}-{\left(\frac{15\pi }{8}\times \frac{I_{Oeff}^{\prime }\sqrt{2V_{in}^{\prime }}}{\mu_{n}C_{ox}\left(W/L\right)}\right)}^{2/5},$$
where *V’_in_*, *V_tn_*, I’_Oeff_, *μ_n_*_,_ and *C_ox_* are an effective voltage amplitude, the threshold voltage of an NMOS transistor, effective loading current, electron mobility, and a gate oxide capacitor, respectively. *W* and *L* are the width and length of the MOSFET, respectively. *I’_Oeff_* and *V’_in_* can be given as follows:(4)$$I_{Oeff}^{\prime }=I_{0}+\frac{I_{S0}}{\pi }\left(\frac{W}{L}\right)\left(1-e^{-V_{in}^{\prime }/V_{T}}\right)\left(1+\lambda_{sub}V_{in}^{\prime }\right),$$
(5)$$V_{in}^{\prime }=\frac{C_{t}}{C_{t}+C_{par}}$$
where *I*_0_, *I*_*S*0_, *V_T_*, *λ_sub_*, *C_t_*, *C_par_*, and *V_in_* are the initial loading current, saturation current, thermal voltage, channel length modulation coefficient, total capacitance value of all capacitors, parasitic capacitance at each stage, and peak amplitude of an input signal of a voltage rectifier, respectively.

If the body effect is ignored, the output voltage of the N-stage rectifier is as follows:(6)$$V_{ON}\left(V\right)=N\times V_{boost}\left(V\right)=N\times \left(V_{in}^{\prime }-V_{tn}-{\left(\frac{15\pi }{8}\times \frac{I_{Oeff}^{\prime }\sqrt{2V_{in}^{\prime }}}{\mu_{n}C_{ox}\left(W/L\right)}\right)}^{2/5}\right),$$

According to the simulation results, many rectifiers are used for achieving the maximum efficiency. For example, 2-, 4-, 6-, 8-, 10-, and 16-stages are designed with respect to different loading currents or peak voltages [22]. For a rapid charging mechanism and stable power source at an operating frequency of 917 MHz, this study adopted a 6-stage voltage rectifier with a small threshold voltage of 0.45 V.

Figure 3 shows a single-ended 6-stage Dickson voltage multiplier, which was published in [22]. It includes 6 diode-connected MOSFETs (M_1_–M_6_) and 6 capacitors (C_1_–C_5_ and C_L_). All transistors and capacitors are identical. A bottom plate, which is marked using a bold line, exhibited a large parasitic capacitance. It is grounded to reduce loss or is connected to the input terminal RF_in_, which is fed from the matching network.

If the input signal of the voltage rectifier was sinusoidal with *RF_in_* = V_in_cos *2**πft*, where V_in_ and *f* are respectively the peak amplitude and operating frequency, the DC output voltage V_dc_ could be obtained for a charge transfer with a load capacitor C_L_. The capacitor is sufficiently large to store the transferred charge and to reduce the output ripple voltage [22].

If the voltage multiplier was in a steady state, RF_in_ was greater than or equal to zero for 0 ≤ t ≤ π/4 and 3π/4 ≤ t ≤ π, and RF_in_ was less than or equal to zero for π/4 ≤ t ≤ 3π/4 in the first time cycle T (= 1/*f*). RF_in_ charged the capacitor C_1_ through the MOSFET M_1_ for RF_in_ ≤ 0, and a steady voltage of *V_in_ − V_TH_* was then stored in C_1_ by reducing a threshold voltage *V_TH_*. Voltage was subsequently changed to *V_in_* + (*V_in_* − *V_TH_*) across the capacitor C_1_ for RF_in_ ≥ 0. M_2_ was the conducting MOSFET, and the voltage of C_2_ was charged to 2 × (*V_in_* − *V_TH_*) by reducing a threshold voltage *V_TH_* of M_2_. In the second time cycle (2T), RF_in_ charged the capacitor C_2_ to *V_in_* + 2 × (*V_in_ − V_TH_*) for RF_in_ ≤ 0, and a steady voltage of 3 × (*V_in_ − V_TH_*) was passed through M_3_ and stored in C_3_. For RF_in_ ≥ 0, the voltage of C_3_ was abruptly changed to *V_in_* + 3 × (*V_in_ − V_TH_*), and M4 was the conducting MOSFET. A steady voltage of 4 × (*V_in_ − V_TH_*) was generated across the capacitor C_4_. After the third time cycle (3T) was completed, a steady voltage of 6 × (*V_in_ − V_TH_*) was generated without a body effect across the load capacitor C_L_. The ideal output DC voltage Vdc could be expressed as follows:(7)$$V_{dc}\left(V\right)=6\times \left(V_{in}-V_{TH}\right),$$

Equation (7) indicates that the larger the threshold voltage *V_TH_* is, the smaller the DC output voltage *V_dc_* is. Threshold voltage *V_TH_* is highly correlational to a semiconductor process. A conventional CMOS has an inherent threshold voltage of approximately 0.45 V for the standard TSMC 0.18 process, whereas a low threshold voltage of 28 mV was obtained using a conventional MOS. The conventional MOS was used to not only develop a new RF-DC rectifier but also to enhance power efficiency, particularly for an ultra-low input power of less than −10 dBm.

### 2.3. Over-Voltage Protection Circuit

Figure 4 presents an over-voltage protection circuit used to prevent the occurrence of a high-voltage breakdown phenomenon during NFC [19]. The output DC voltage V_dc_ of the RF-DC rectifier was connected to the proposed protection circuit, and two PMOSs (M_1_ – M_2_) and two NMOSs (M_8_ – M_9_) were diode-connected. Two bias voltages *V_A_* and *V_B_* could be expressed as follows:(8)$$V_{A}=V_{dc}-\left(\left|V_{OD1}\right|+\left|V_{tp1}\right|\right)-\left(\left|V_{OD2}\right|+\left|V_{tp2}\right|\right),$$
(9)$$V_{B}=V_{OD9}+V_{tn9},$$
where *V**_ODi_**, V_tpi_*, and *V_tni_* are the overdrive voltage of the *i*^th^ MOSFET, threshold voltage of the *i*^th^ PMOS, and threshold voltage of the *i*^th^ NMOS. Under a common-mode operation, the two bias voltages were identical (i.e., *V_A_* = *V_B_*). The DC output voltage of the RF-DC rectifier can be expressed as follows:(10)$$V_{dc}=\left(\left|V_{OD1}\right|+\left|V_{tp1}\right|\right)-\left(\left|V_{OD2}\right|+\left|V_{tp2}\right|\right)+\left(V_{OD9}+V_{tn9}\right),$$

Furthermore, the two bias voltages *V_A_* and *V_B_* could be derived from resistors (*R*_1_ − *R*_2_, *R_ds1_ − R_ds2_*, and *R_ds8_ − R_ds9_*). Thus, the following can be derived:(11)$$V_{A}=\frac{R_{1}}{R_{ds1}+R_{ds2}+R_{1}}\times V_{dc},$$
(12)$$V_{B}=\frac{R_{ds9}}{R_{2}+R_{ds8}+R_{ds9}}\times V_{dc},$$
where *R_dsi_* was the conduction impedance of the *i*^th^ diode-connected MOSFET, which is expressed as (*g_mi_ + g_mbi_*)^−1^ with the transistor transconductance *g_mi_* and body effect transconductance *g_mbi_*. *R*_1_ and *R*_2_ are constant resistors. The larger the bias current *I_Ri_* of the *i*^th^ resistor is, the larger the transconductance *g_mi_* is and the smaller the conduction resistor *R_dsi_* is.

If the output DC voltage V_dc_ of the RF-DC rectifier was higher than the aforementioned voltage [Equation (11)], the bias voltage *V_A_* was larger than the bias voltage *V_B_* because of the reduced conduction impedance *R_dsi_* with the large resistor current *I_Ri_*. The difference between *V_A_* and *V_B_* was amplified, and the differential output voltage V_O_ was used to control the conduction current I_O_ of the output transistor M_O_. The larger the dc voltage V_dc_ is, the larger the conduction current I_O_ is. Thus, a stable DC output voltage ranged from 1.69 V to 1.76 V with an input power (*P_in_*) ranging from −14 dBm to +10 dBm. Please note that the voltage variations of two biased voltages, *V_A_* and *V_B_*, will be suppressed in the PVT (process, supply voltage, and temperature) variation. For example, if the resistor *R*_1_ is reduced with PVT variation, the bias current *I*_*R*1_ is enlarged. Then both conduction resistors, *R_ds1_* and *R_ds2_*, are decreased by the large bias current *I*_*R*1_. As a result, three resistors, *R_ds1_, R_ds2_* and *R*_1_, are reduced simultaneously to suppress the impact of PVT variation.

### 2.4. Low-Voltage Low-Dropout Regulator

Because of an input voltage with inference and input power limitation, a low-voltage and low-power LDO regulator is required for supplying a stable and clean voltage to the next stage. The designed LDO regulator was used for regulating the output variation in the over-voltage protection circuit and for providing a stable voltage V_dd_ of 1.5 V to the charger control circuit. Figure 5 shows the proposed low-voltage LDO regulator, which includes a reference voltage, a current source, an error amplifier, a pass transistor, a feedback network, frequency compensation, and a load. A CMOS reference voltage V_ref_ was generated and inputted to the positive terminal (+) of an error amplifier (EA). The negative terminal (−) of the EA was connected to the feedback network. A comparison between the reference voltage V_ref_ and a feedback value indicates that the voltage difference between the positive and negative terminals was amplified as an output voltage of the EA, which was connected to a pass transistor for providing a stable supply voltage V_dd_ by controlling the load current flow through the pass transistor [23].

Figure 6 shows the complete circuit of the adopted CMOS reference voltage [24]. The supplied voltage V_dc_ could be maintained at a possible voltage of 0.7 V when all transistors operated in the subthreshold region. A compensation capacitor C_C_ was added between the drain of M_N1_ and ground (GND) to improve circuit stability by increasing the phase margin. Assume that I_2_ = 100 × I_1_, W_MP2_ = 100 × W_MP1_, and W_MN2_ = 100 × W_MN1_. The voltage of node X was equal to that of node Y (i.e., V_X_ = V_Y_). Thus, the drain–source voltage of M_P2_ (V_DS,MP2_) was equal to that of M_P3_ (V_DS,MP3_). Two bias currents (I_2_ and I_3_) were identical without channel length modulations of M_P2_ and M_P3_. If two NMOSFETs operated in the subthreshold region, the voltage difference across the resistor *R*_1_ is given as follows:(13)$$V_{R1}=V_{GS,MN2}-V_{GS,MN3}=nV_{T}\times \ln\frac{{\left(W/L\right)}_{MN3}}{{\left(W/L\right)}_{MN2}},$$
where *n* and *V_T_* are the subthreshold region swing parameter and thermal voltage, respectively. The *n* is a constant and *V_T_* is a parameter independent of the process. Thus, process variations do not influence *V*_*R*1_. *V_GS, MNi_* and *(W/L)_MNi_* are the gate–source voltage and the ratio of width to length for the transistor M_Ni_, respectively. Equation (13) indicates that a temperature coefficient (TC) is positive for the resistor’s voltage *V*_*R*1_.

Voltage reference *V_ref_* can be written as follows:(14)$$V_{ref}=V_{R2}+V_{GS,MN2}=2.01\times \frac{R_{2}}{R_{1}}\times V_{R1}+V_{GS,MN2},$$
where *V*_*R*1_ and *V*_*R*2_ are voltage differences across resistors *R*_1_ and *R*_2_, respectively, and *V_GS,MN2_* is the gate–source voltage of the transistor *M*_*N*2_ with a negative TC. A positive TC was combined with a negative TC in an integrated circuit to obtain the desired reference voltage *V_ref_* with zero temperature dependence [24]. The simulation result shows that the reference voltage *V_ref_* varied from 499.035 mV to 502. 855 mV with respect to the supplied voltage V_dc_ from 1.7 V to 2.0 V at a quiescent current of 34 nA.

Figure 7 shows the complete circuit of a low-voltage LDO regulator. The EA was a CMOS two-stage amplifier, which was designed to regulate the supplied output voltage V_dc_ of the low-voltage LDO regulator by controlling the gate voltage of the pass transistor V_err_. The first stage had a p-channel differential input pair with an n-channel current mirror active load for a high gain *A_V1_*. The second stage is generally configured as a simple common source stage to allow maximum output swings with gain *A_V2_* [25]. Bias currents were copied through the current source by controlling the resistor *R*_3_. The voltage swing at V_err_ was equal to V_dc_ − |V_OD,MP6_| − V_OD,MN6_ with overdrive voltages, V_OD,MP6_ and V_OD,MN6_, of MP6 and MN6, respectively. The overall voltage gain A_V_ can be derived as follows:(15)$$\{\begin{array}{c} A_{v1}=-g_{m,MP7}\left(r_{ds,MP8}||r_{ds,MN5}\right)\\ A_{v2}=-g_{m,MN6}\left(r_{ds,MP6}||r_{ds,MN6}\right)\\ A_{v}=A_{v1}\times A_{v2} \end{array}$$
where *g_m,MPi_* and *r_ds,MPi_* are the transconductance and conduction resistance of the *i*^th^ PMOS, respectively. *g_m,M_**_Ni_* and *r_ds,M_**_Ni_* are the transconductance and conduction resistance of the *i*^th^ NMOS, respectively.

Zero frequency ω_z_ can be modified by placing a resistor *R_z_* in series with the compensation capacitor *C_Z_* [26]. If *R_z_* ≥ (*g_m,MN6_*)^−1^, then *ω_z_* ≤ 0. Zero may be moved into the left-half plane for cancelling the first nondominant pole ω_p2_. The compensation resistance *R_Z_* is then given as follows:(16)$$R_{z}=\frac{1}{g_{m,MN6}}\left(1+\frac{C_{1}+C_{2}}{C_{Z}}\right),$$
where *C_1_* and *C_2_* are total capacitances at node E before *C_Z_* was added and that at node V_err_, respectively. Moreover, the increasing *C_Z_* moves the dominant pole to a lower frequency without affecting the second pole. This effect ensures that the designed amplifier is more stable [25]. The simulated results indicate that DC gain, phase margin, and unit-gain bandwidth were 50 dB, 60°, and 1.57 MHz, respectively, at a supplied voltage of 1.7 V and quiescent current of 223.22 nA.

The feedback network was a voltage-trimming network and comprised four diode-connected PMOSs (M_P10_–M_P13_) and an adjustable voltage, *V_A_*, which is generally connected to a voltage of 1.4 V. Three PMOSs were designed to complete coarse adjustment, whereas the NMOS M_P10_ was used to perform fine-tuning by controlling the adjustable voltage *V_A_*. To stabilize the LDO regulator, phase characteristics were adjusted such that a phase shift was less than 180° at gain crossover. Frequency compensation is completed by connecting a resistor *R_F_* in series with a filter capacitor *C_F_*. The loop gain was zero at *s_z_* = *−(R_F_C_F_)^−^*^1^.

### 2.5. Charge Control Circuit

Figure 8 presents the charge control circuit, which was used to control the charge in the battery by using the supplied output voltage V_dd_ of the LDO regulator. The control circuit comprised a differential pair (Q_1_–Q_3_), current mirror (Q_4_–Q_5_), and comparator (Comp) to prevent the overcharging of the battery. Channel length modulation and body effect were assumed negligible, and the differential pair operated in a saturation region. The battery was charged using a constant current, which was generated using Q_1_ and Q_2_ with two bias voltages (V_B1_ and V_B2_). When the battery voltage V_bat_ was lower than the reference voltage V_ref_, the output voltage of the comparator increased to a high level (1) and Q_3_ was turned off. Two constant currents of Q_1_ and Q_4_ simultaneously flowed into Q_2_. The conduction current of Q_5_ was amplified N times with (W/L)_5_ = N×(W/L)_4_ when it was passed through the current mirror Q_4_–Q_5._ This large current rapidly charged the battery. If V_bat_ was higher than V_ref_, the output voltage of the comparator was low (0) and Q_3_ was turned off. Two constant currents of Q_1_ and Q_3_ simultaneously flowed into Q_2_. Moreover, Q_4_ and Q_5_ were turned off, and the battery was not further charged.

## 3. Simulated and Measured Results

Figure 9 presents the simulated output voltage of the RF-DC rectifier by using a signal analyzer (EXA N9010A) at a distance of 10.0 cm between the antenna and rectifier. The operating frequency was 915 MHz, and the equivalent load was 1 MΩ. The simulation results revealed a minimum output voltage of 0.746 V at an input power *P_in_* of −20 dBm, and a maximum output voltage of 21.693 V at *P_in_* = +20 dBm. The proposed RF-energy-harvesting chip requires the over-voltage protection circuit to prevent the breakdown phenomenon, which is generated by high output voltages, particularly for NFC. Figure 10 shows the simulated DC output voltages of the RF-DC rectifier with over-voltage protection, which were limited from 1.773 V to 1.809 V with respect to the input power *P_in_* from −13 dBm to +20 dBm. Figure 11 presents the simulated *PCE* of the RF-DC rectifier with over-voltage protection, which is termed the rectifier *PCE*. It was approximately 43.601% at an input power of −7 dBm. *PCE* is defined as that in [26]:(17)$$PCE(\%)=\frac{P_{dc}}{P_{in}}\times 100\%=\frac{I_{dc}\times V_{dc}}{P_{in}}\times 100\%$$

The proposed LDO regulator provides a stable output voltage of 1.5 V and a load current of 30 μA to the next stage-charge control circuit. When the input voltage varied from 1.7 V to 2.0 V and the reference voltage was 1.4 V, the LDO regulator performed with a quiescent current of 317 nA, and a maximum power efficiency of 84.835% was obtained at an input power *P_in_* of −10 dBm. Figure 12 shows the simulated line regulation of the proposed LDO regulator. When the input voltage was varied from 1.7 V to 2.0 V, the output voltage changed from 1.5 V to 1.5108 V, and a line regulation of 36 mV/V was obtained. Figure 13 presents the simulated load regulation of the proposed LDO regulator. As the output current varied from 0.0 μA to 10.0 μA, the output voltage changed from 1.499952 V to 1.499968 V. A load regulation of 1.6 mV/mA was then obtained [27].

The charge control circuit was inputted with a pulse waveform, which is gradually ramped up from 1.2 V to 1.5 V, to charge the battery with the charging current of 16 μA. When the battery voltage was charged to 1.4 V at 2.002 ms, the charging current decreased sharply, thereby reducing the charging speed. Finally, the voltage of the battery increased to 1.50 V with zero charging current. Figure 14 shows the simulated battery voltage and charging current of the charger control circuit. The maximum power efficiency of the charge control circuit was 84.835% at an input power *P_in_* of +20 dBm.

When all functional blocks were verified, the RF-energy-harvesting chip could be implemented using the standard TSMC 0.18 μm 1P6M CMOS process. Figure 15 shows the layout of the proposed RF-energy-harvesting IC, which comprised the RF-DC rectifier, over-voltage protection circuit, CMOS voltage reference, LDO regulator, and charge control circuit. The over-voltage protection circuit, low-voltage LDO regulator, and charge control circuit of the RF-DC rectifier required 33 μW, 5 μW, and 4 μW, respectively, to charge the battery to 24 μW. The simulation results revealed that charging current was 16 μA at N = 55, and the over-voltage protection mechanism was started at 2.002 by setting the output voltage of the over-voltage protection circuit, output voltage of the charger control circuit, and reference voltage at 1.7 V, 1.5 V, and 1.4 V, respectively. Figure 16 shows the simulated total power conversion efficiency of the proposed RF-energy-harvesting IC, which was termed the system *PCE*. The maximum system *PCE* was 29.873% at an input power *P_in_* of −12 dBm. The maximum rectifier *PCE* was reduced to the same value at *P_in_* = −12 dBm by integrating all designed circuits into a single chip (Figure 11). Table 1 summarizes the performance and compares it with that of other RF-DC rectifiers. The simulated maximum rectifier *PCE* in this study was superior to the *PCEs* of previously published RF-DC rectifiers. Table 2 presents the simulated specifications of the proposed RF-energy-harvesting IC.

The matching network was completed with off-chip components to have an input impedance of approximately 50 Ω. Figure 17 shows the matching network, which was designed to obtain not only maximum power transfer but also maximum conversion efficiency. The E5071C network analyzer was used to measure the input impedance against various input powers. Moreover, multiple sets of matching circuits were designed to ensure that each set of the matching circuit exhibited an input return loss (S_11_) of less than −25 dB. For example, an input impedance of 51.464 – j × 1.3071 could be achieved with an input return loss (S_11_) of −34.271 dB by putting a parallel inductance (L) of 7.5 nH and a series capacitance (c) of 22 pF on a printed circuit board (PCB). Figure 18 shows the measured input return loss in the Z-smith chart at an input power and RF frequency of 0 dBm and 915.0000 MHz, respectively.

Figure 19 shows the measurement platform of the designed RF-energy-harvesting IC with capacitor, which includes the PCB (15.576 × 7.906 cm^2^), network analyzer (Agilent E5071C), RF-IC (0.58 × 0.49 mm^2^), Powercast transmitter (TX91501), and multimeter (CNN 38). By connecting the receiving antenna to the matching network on the measured PCB, the multimeter measures the output voltage of approximately 1.702 V by setting a distance of 2.45 m between the Powercast transmitter and PCB, with an input power of 0.0 dBm. Figure 20 shows the measured setup of the proposed RF-energy-harvesting IC.

Table 3 summarizes the results for the designed RF-DC rectifier with the over-voltage protection circuit. Because the distance between the Powercast transmitter and test PCB was provided, the input power *P_in_* (dBm), input return loss *S_11_* (dB), and DC output voltage V_dc_ (V) were measured after the input impedance *Z_in_* (Ω) and its corresponding matching circuits [L (nH) and C (pF)] were completed. The results showed that the DC output voltage V_dc_ varied from 0.291 V to 1.725 V with respect to the distance from 5.45 m to 0.55 m. The measured output voltages were lower than the simulated results. A stable charging voltage V_dd_ can be obtained through the LDO regulator. Figure 21 shows the measured DC output voltage Vdc of the designed RF-DC rectifier with the over-voltage protection circuit with respect to the input power *P_in_* from −20 dBm to +10 dBm. Please note that the measured DC output voltages were limited from 1.682 V to 1.725 V with respect to the input power from −2 dBm to +10 dBm in Figure 21.

Finally, the simulated rectifier *PCE* was 43.601% at an input power *P_in_* of −7 dBm, and the simulated system *PCE* was 29.873% at an input power *P_in_* and a load resistor of −12 dBm and 1 MΩ, respectively. The limitation of system integration is the reduction of the* PCE* from 43.601% to 29.873%. The improvement of the system *PCE* is a crucial concern. This study focused on increasing the system *PCE* by changing the load resistor from 20 kΩ to 38 kΩ. Figure 22 exhibits the measured system* PCE* with respect to the input power *P_in_* (dBm) and load resistor. The larger the load resistor is, the smaller the input power is. A large load resistor reduces load current, whereas it increases charging current I_bat_ for the abrupt charging of the battery. This phenomenon provides the maximum system *PCE* at a low input power. A trade-off was observed between the load resistor and the maximum system *PCE*. Moreover, the results indicate that the largest system *PCE* of 40.556% was observed at a load resistor of 30 kΩ and an input power *P_in_* of −6 dBm.

Table 4 shows the measured electrical characterizations of the proposed RF-energy-harvesting IC, which includes the power consumption, power delivery, and other electrical characterization facts. Table 5 summarizes the performance and compares the performance with that of other RF-DC rectifiers and RF-energy-harvesting ICs. The maximum rectifier *PCE* obtained in this study was superior to that in [30], and the maximum *PCE* of the RF-energy-harvesting IC was superior to that in [31]. Several RF-DC rectifiers have been studied in the past few years; however, studies on system integration, such as RF-energy-harvesting ICs, have been rare. Furthermore, the IC proposed in this study exhibited low power consumption and a small chip area. It is possible to decrease the labor cost significantly by eliminating the future maintenance efforts to replace batteries. At close range, this proposed IC can be used to trickle charge for low power devices including GPS, tracking tags, wearable sensors, and consumer electronics. At long range, this transmitted power can be used for battery-based or battery-free remote sensors for factory automation, structural health monitoring, and industrial control. In future work, the power conversion efficiencies (*PCEs*) of the rectifier and RF-energy harvesting chip can be improved by reducing the energy consumption of each proposed circuit.

## 4. Conclusions

This study proposed an auxiliary power integrated chip (IC) to supply power to the WSN with a Power cast transmitter of ISM 915 MHz. The RF-energy-harvesting IC was designed and fabricated using the standard TSMC 0.18 μm 1P6M CMOS process. The externally matched capacitors and inductors were manufactured in the matching network by MuRata Company. On integrating the externally matched components with the designed RF-energy-harvesting IC, the simulated results showed that the maximum *PCEs* of the rectifier and harvesting IC were 43.6% and 29.873%, respectively, at an ISM band of 915 MHz, an input power of −7 dBm, and a load of 1 MΩ. The output voltage of the RF-DC rectifier with the over-voltage protection circuit was limited from 1.773 V to 1.809 V with the varying input power from −10 dBm to +20 dBm. A stable voltage of 1.5 V was supplied to the charge control circuit passing through the LDO circuit. Measurements validated that the designed RF-energy-harvesting IC works successfully. The output voltage V_dc_ varied from 0.291 V to 1.725 V with respect to the distance from 5.45 m to 0.55 m. The large load resistor reduced load current; however, it sharply increased the charging current to charge the battery abruptly. This phenomenon provided the maximum system *PCE* at low input power. A trade-off was observed between the load resistor and the maximum system *PCE*. Furthermore, the maximum rectifier *PCE* of this study was superior to that in [30], and the maximum *PCE* of the RF-energy-harvesting IC was superior to that in [31]. Measurements indicate that the IC used in this study exhibited low power consumption and a small chip area. The proposed energy harvesting IC can be used in both ambient source and dedicated source [34]. Another possibility offered by the use of microelectronic substrates is on-chip photovoltaic generation with integrated photodiodes [35,36].

## Figures and Tables

**Figure 1 sensors-19-01754-f001:**
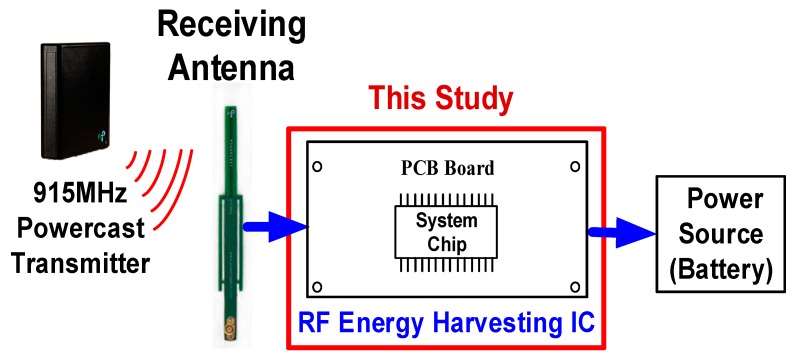
Proposed RF-energy-harvesting system.

**Figure 2 sensors-19-01754-f002:**
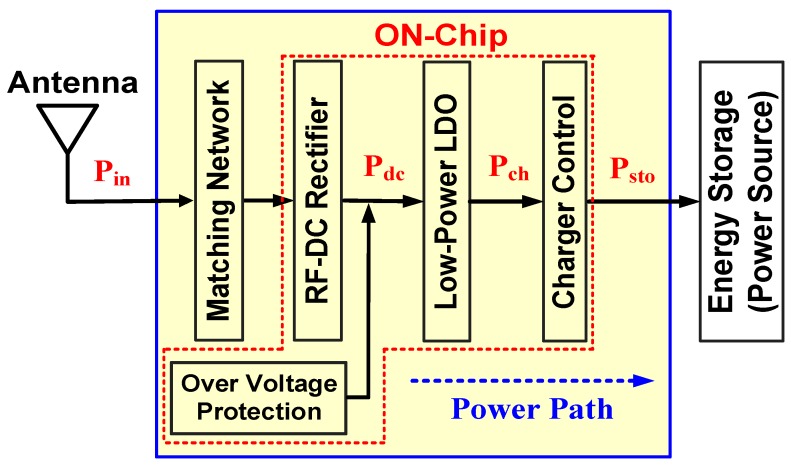
Functional diagram of RF-energy-harvesting IC.

**Figure 3 sensors-19-01754-f003:**
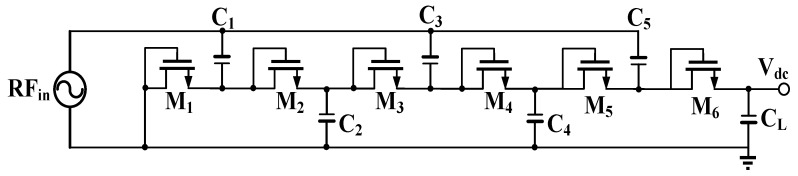
Six-stage Dickson voltage multiplier circuit.

**Figure 4 sensors-19-01754-f004:**
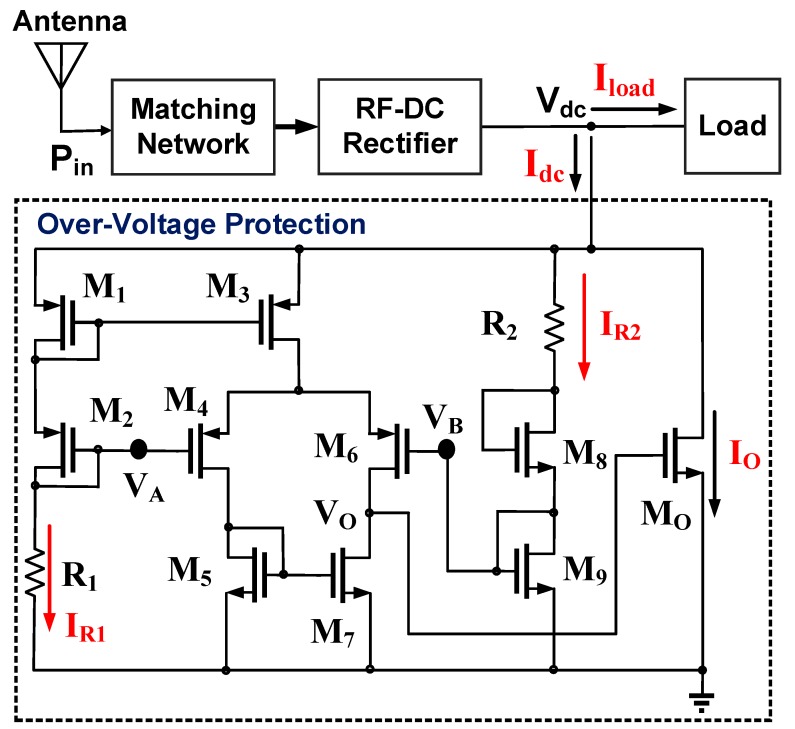
Proposed over-voltage protection circuit.

**Figure 5 sensors-19-01754-f005:**
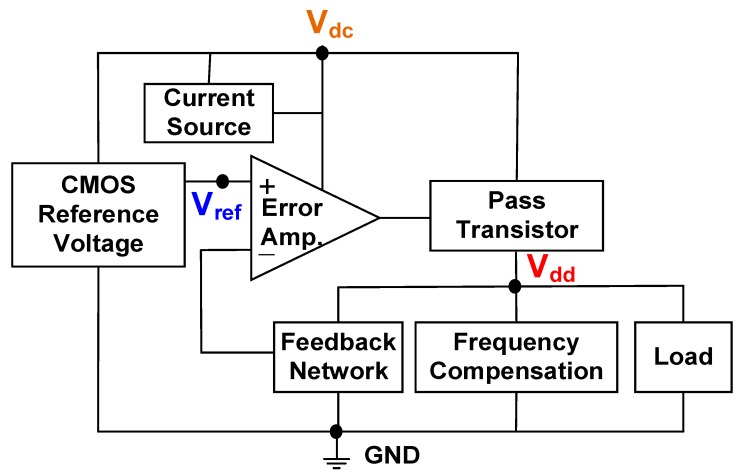
Proposed low-voltage LDO regulator with a reference voltage, a current source, an error amplifier, a pass transistor, a feedback network, frequency compensation, and a load.

**Figure 6 sensors-19-01754-f006:**
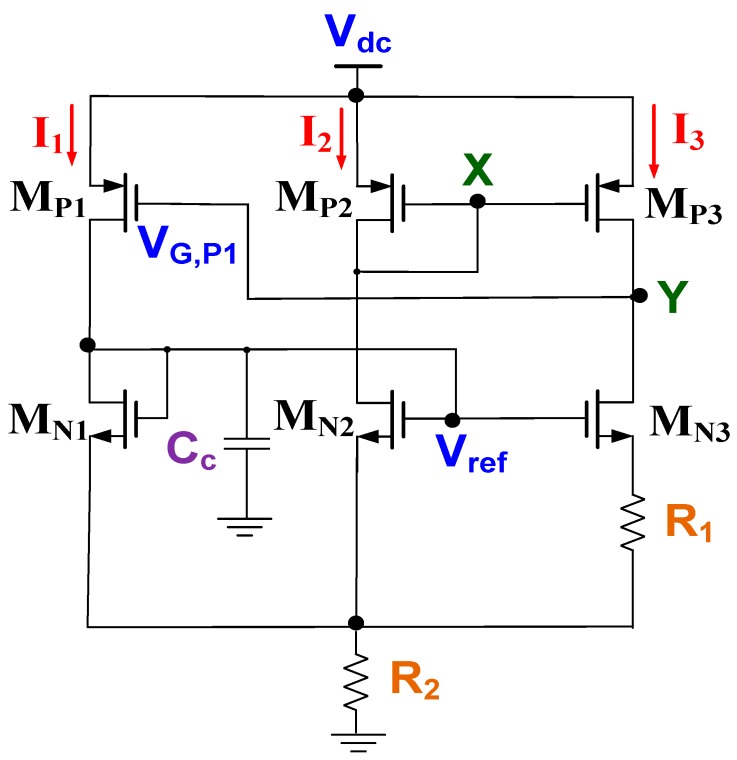
Complete circuit of the adopted CMOS reference voltage.

**Figure 7 sensors-19-01754-f007:**
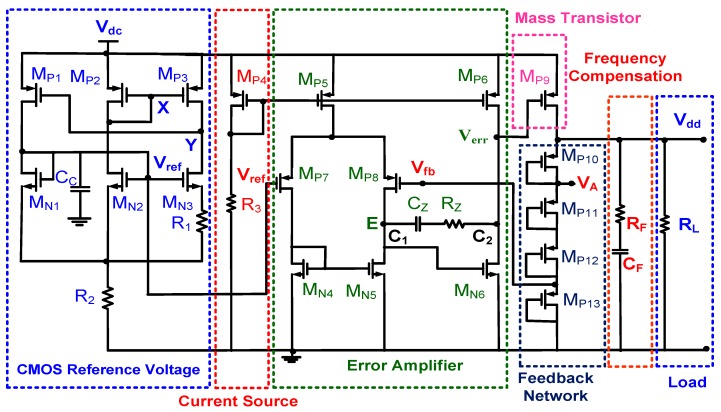
Complete circuit of the low-voltage LDO regulator.

**Figure 8 sensors-19-01754-f008:**
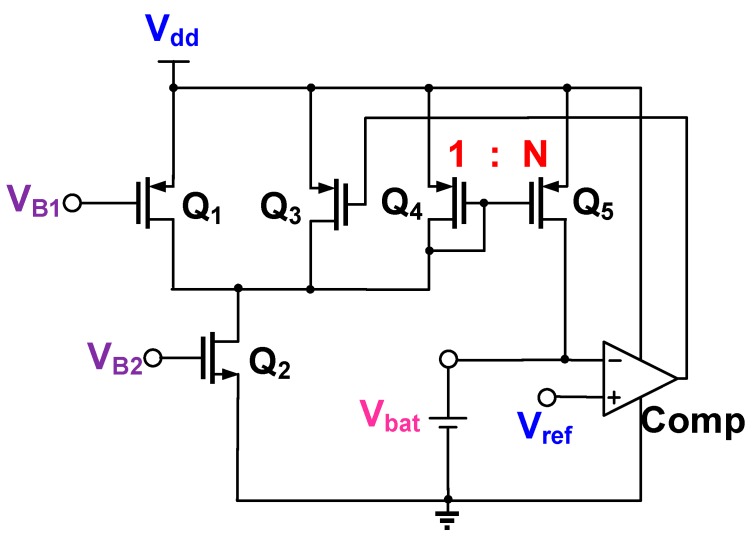
Complete circuit of charge control circuit.

**Figure 9 sensors-19-01754-f009:**
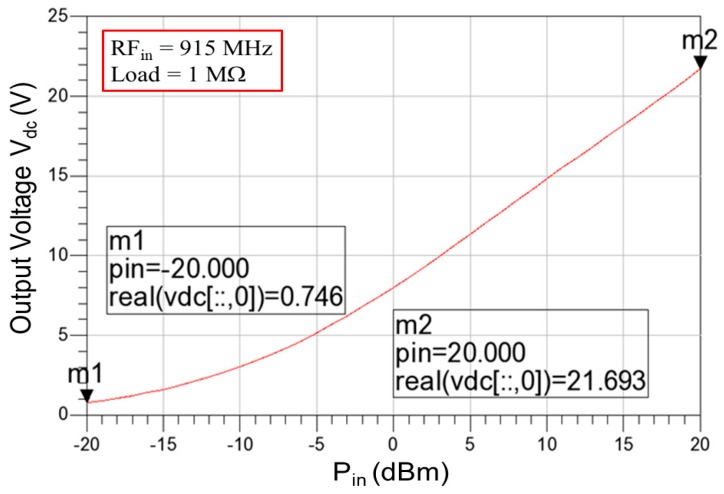
Simulated output voltage of the RF-DC rectifier with respect to the input power from −20 dBm to +20 dBm.

**Figure 10 sensors-19-01754-f010:**
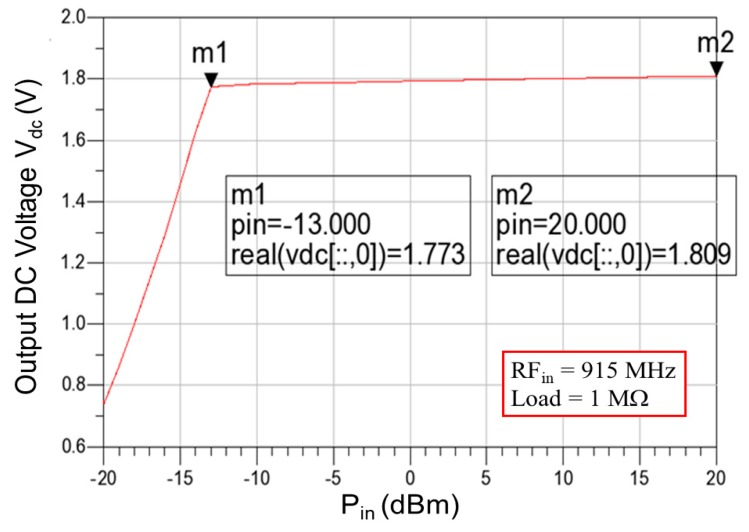
Simulated DC output voltages of the RF-DC rectifier with over-voltage protection.

**Figure 11 sensors-19-01754-f011:**
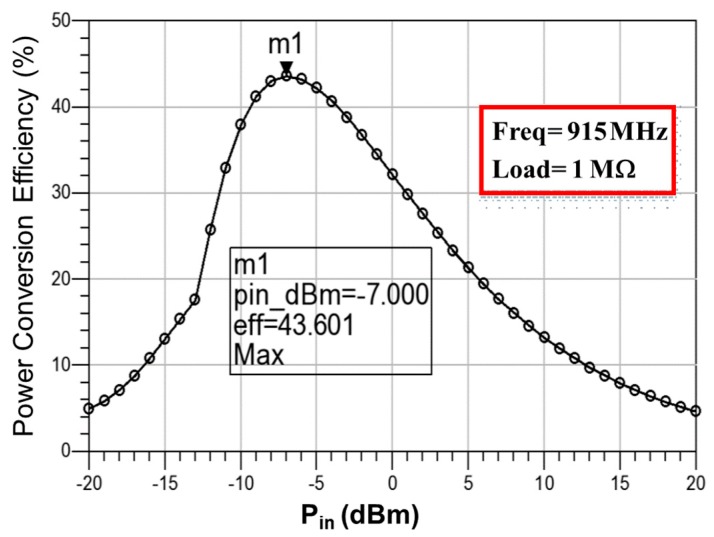
Simulated *PCE* of the RF-DC rectifier with over-voltage protection (rectifier *PCE*), whose *PCE* was approximately 43.6% at the input power of −7 dBm.

**Figure 12 sensors-19-01754-f012:**
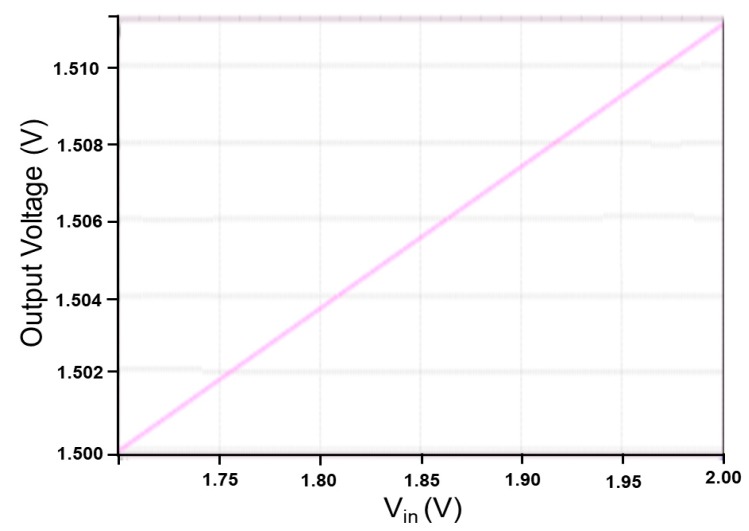
Simulated line regulation of the proposed LDO regulator.

**Figure 13 sensors-19-01754-f013:**
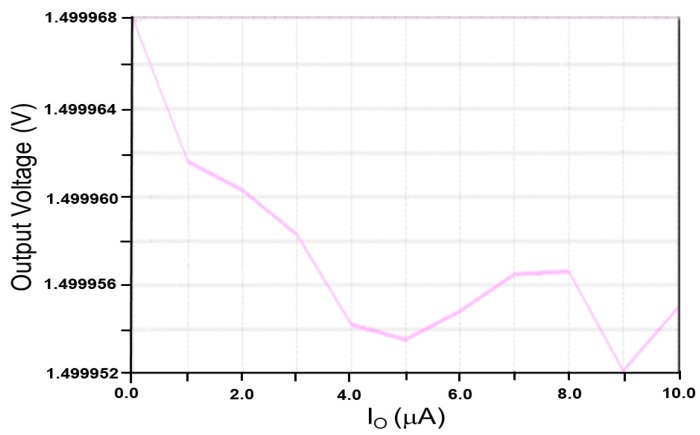
Simulated load regulation of the proposed LDO regulator.

**Figure 14 sensors-19-01754-f014:**
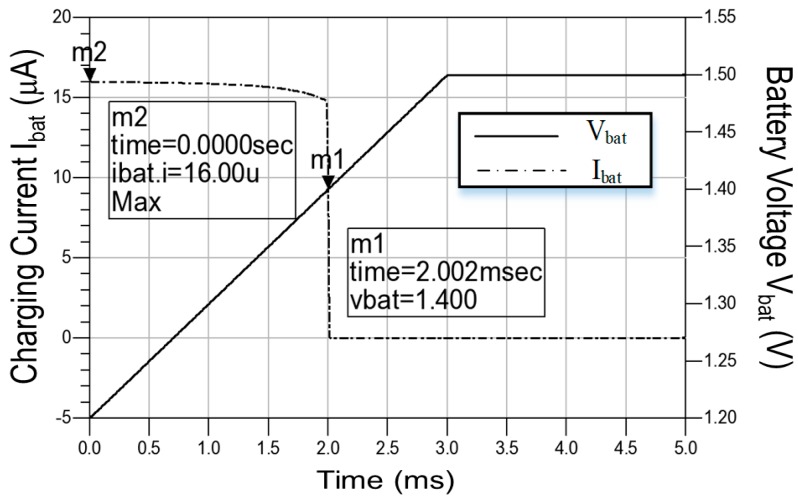
Simulated battery voltage and charging current of the charger control circuit.

**Figure 15 sensors-19-01754-f015:**
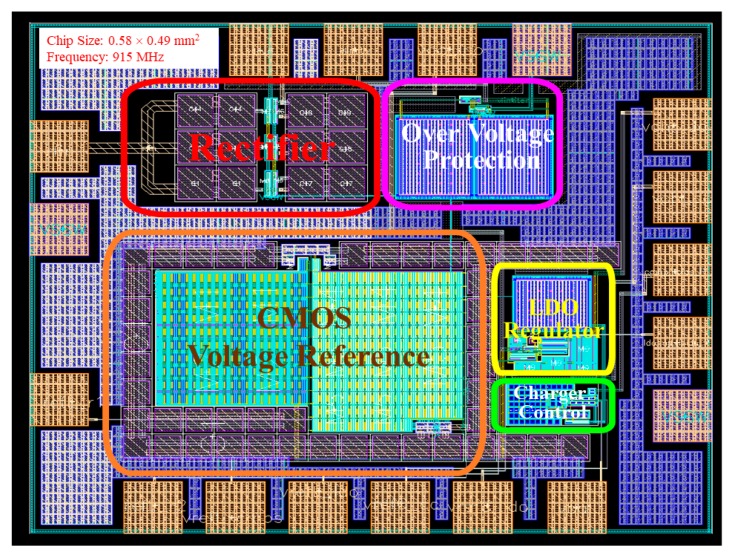
Layout of the proposed RF-energy-harvesting IC.

**Figure 16 sensors-19-01754-f016:**
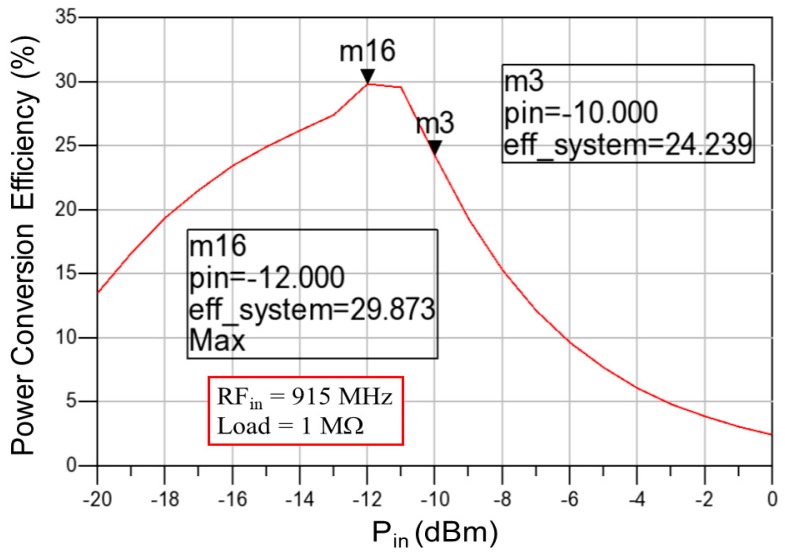
Simulated total *PCE* of the proposed RF-energy-harvesting IC.

**Figure 17 sensors-19-01754-f017:**
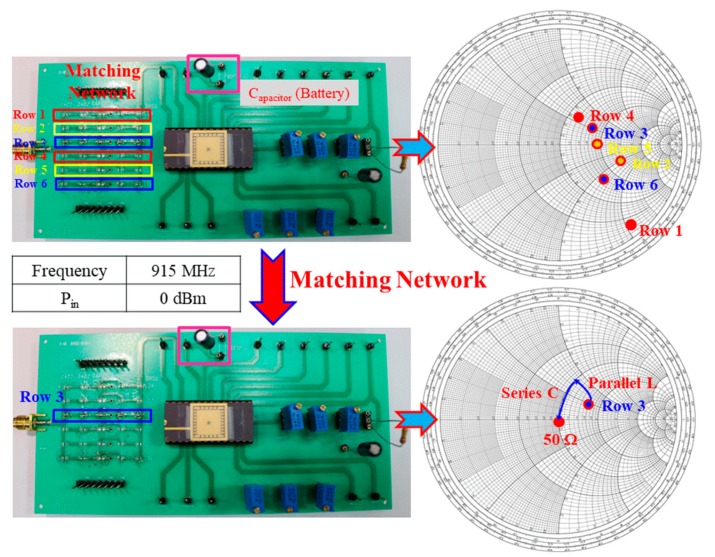
Matching network used to obtain an input impedance of approximately 50 Ω.

**Figure 18 sensors-19-01754-f018:**
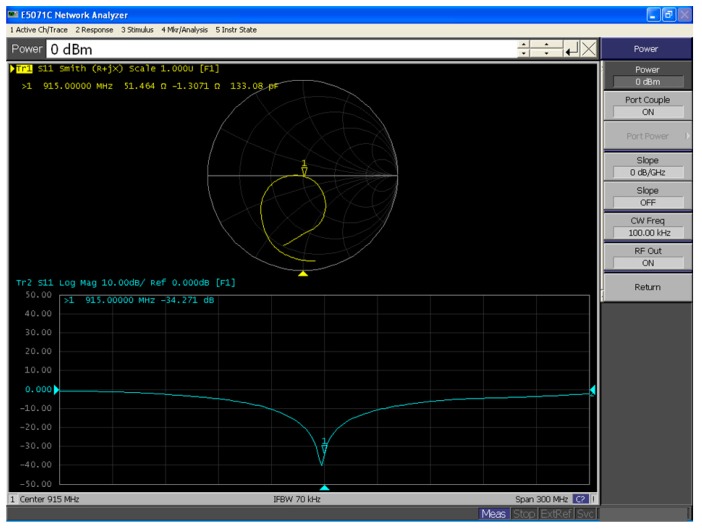
Measured results in the Z-smith chart at an input power of 0 dBm and RF of 915.0000 MHz.

**Figure 19 sensors-19-01754-f019:**
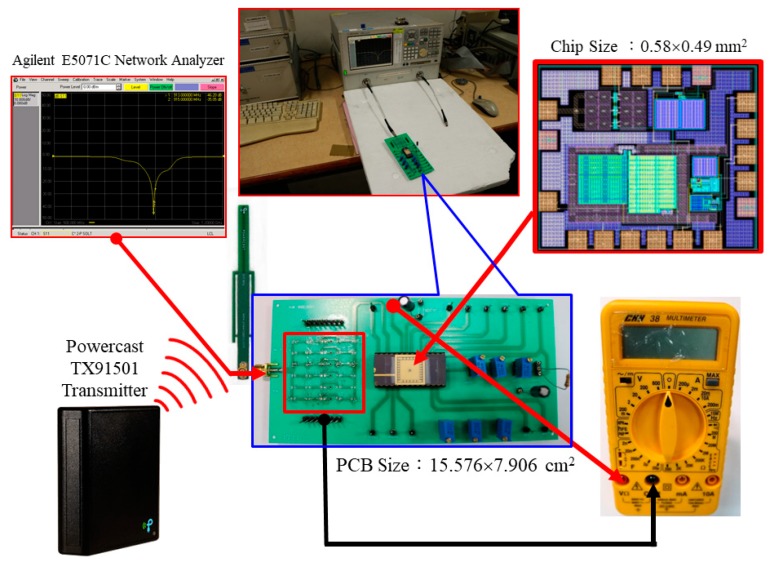
Measurement platform of the designed RF-energy-harvesting IC with capacitor.

**Figure 20 sensors-19-01754-f020:**
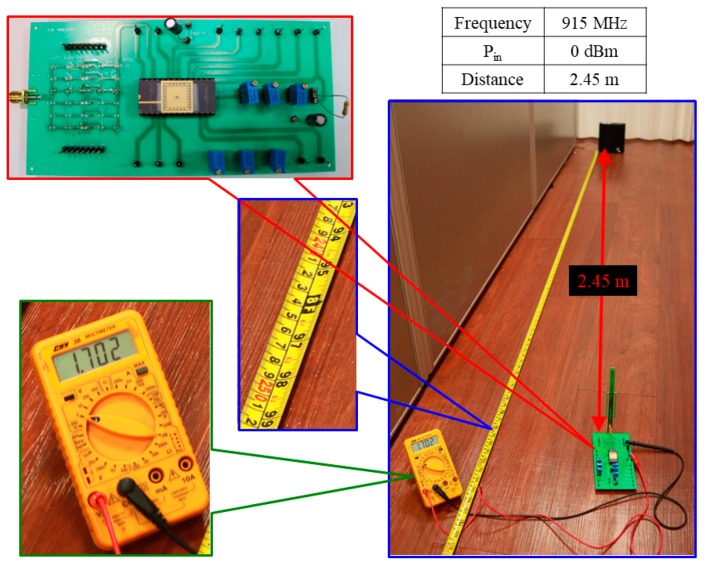
Measured setup of the proposed RF-energy-harvesting IC.

**Figure 21 sensors-19-01754-f021:**
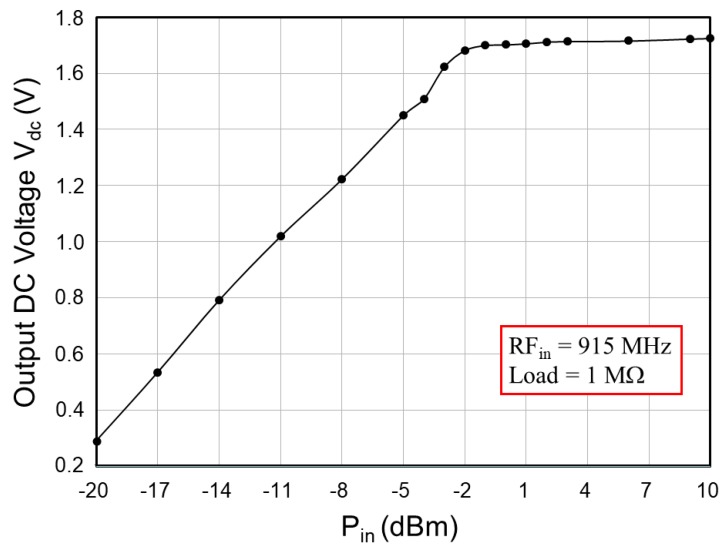
Measured DC output voltages of the RF-DC rectifier with over-voltage protection with respect to the input power.

**Figure 22 sensors-19-01754-f022:**
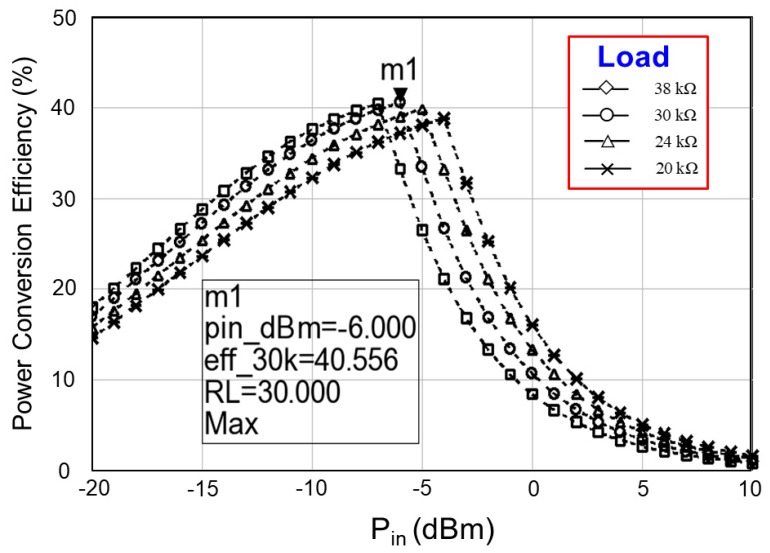
Measured *PCE* of the proposed RF-energy-harvesting IC (system *PCE*) with respect to the input power (dBm) and load resistor (Ω).

**Table 1 sensors-19-01754-t001:** Summary of performance and its comparison with those of previous RF-DC rectifiers.

Reference (year)	[19] 2013	[28] (2014)	[29] (2017)	This Work
Process	0.18 μm	0.18 μm	65 nm	0.18 μm
Input power	−7 dBm	−21.2 dBm	−17.7 dBm	−7 dBm
RF frequency	5.2 GHz	925 MHz	900 MHz	915 MHz
Rectifier stages	5	7	5	6
Matching circuit	On-chip	Off-chip	Off-chip	Off-chip
Maximum *PCE*	42%	43%	36.5%	43.6%

**Table 2 sensors-19-01754-t002:** Simulated specifications of the proposed RF-energy-harvesting IC.

Parameters	Designed Specifications
Technology process	TSMC 0.18 μm 1P6M CMOS process
RF frequency (*f*_in_)	915 MHz
Input power range (*P_in_*)	−13 ~ +20 dBm
Output dc voltage (V_dc_)	1.773 ~ 1.809 V
Maximum *PCE* of rectifier	43.603%
LDO output voltage (V_dd_)	1.50 V
Charging current (I_bat_)	16 μA
Maximum *PCE* of RF IC	29.873%
Chip area (include pads)	0.58 × 0.49 mm^2^

**Table 3 sensors-19-01754-t003:** Measured input power *P_in_* (dBm), input return loss *S_11_* (dB), and output DC voltage V_dc_ (V) with respect to distance (m) between the transmitter and test PCB, input impedance *Z_in_* (Ω) and its corresponding matching circuits [L (nH) and C (pF)] for the RF-DC rectifier with the over-voltage protection circuit.

Distance (m)	*Z_in_* (Ω)	Matching Circuits		*P_in_* (dBm)	*S_11_* (dB)	V_dc_ (V)
		Parallel (L)	Series (C)			
5.45	50.548 − j × 2.294	2.4 nH	38 pF	−20	−28.1	0.291
5.12	52.043 − j × 2.893	3.6 nH	32 pF	−17	−28.4	0.535
4.66	53.916 − j × 1.260	6.8 nH	29 pF	−14	−32.6	0.792
4.25	51.784 − j × 2.789	8.6 nH	26 pF	−11	−29.8	1.019
3.93	52.784 − j × 2.699	6.6 nH	24 pF	−8	−28.4	1.223
3.60	48.561 − j × 6.422	5.6 nH	22 pF	−5	−33.0	1.450
3.30	52.109 + j × 3.382	5.6 nH	22 pF	−4	−28.3	1.509
3.15	53.942 − j × 1.324	6.2 nH	22 pF	−3	−28.0	1.623
2.90	54.226 + j × 2.072	6.2 nH	22 pF	−2	−26.9	1.682
2.70	53.369 − j × 11.324	6.8 nH	22 pF	−1	−29.0	1.700
2.45	51.464 − j × 1.307	7.5 nH	22 pF	0	−34.3	1.702
2.25	51.581 + j × 7.158	10 nH	10 pF	1	−35.4	1.706
1.68	49.710 − j × 4.909	10 nH	10 pF	2	−26.1	1.712
1.50	53.573 + j × 1.835	12 nH	10 pF	3	−28.2	1.714
1.00	54.669 − j × 1.756	22 nH	10 pF	6	−26.4	1.716
0.68	53.350 − j × 2.136	68 nH	10 pF	9	−28.3	1.723
0.55	50.655 − j × 4.093	100 nH	10 pF	10	−27.7	1.725

**Table 4 sensors-19-01754-t004:** Measured electrical characterizations of the proposed RF-energy-harvesting IC.

Parameters (unit)	Values
RF frequency (MHz)	915
Input impedance (Ω)	51.464 − j × 1.3071
Input power (dBm)	0
Distance between Powercastand PCB (m)	2.45
Matching components	L=7.5 nH and C=22 pF
Load resistor (kΩ)	30
Measured PCB (cm^2^)	15.576 × 7.906
Maximum *PCE*	40.566% @*P_in_* = −6 dBm
Output voltage (V)	1.50
Power consumption (μW)	42
Chip area (mm^2^)	0.58 × 0.49

**Table 5 sensors-19-01754-t005:** Summary of performance and its comparison with those of other RF-DC rectifiers and RF-energy-harvesting ICs.

Reference (year)	[30] (2018)	[31] (2018)	[32] (2016)	[33] (2017)	This Work
Process (μm)	0.13	0.18	0.18	0.13	0.18
RF frequency (MHz)	915	900	900	2000	915
Matching circuits	Off-chip	Off-chip	Off-chip	Off-chip	Off-chip
Input power range (dBm)	−35 ~ −15	−30 ~ +0	−26 ~ −8	−35 ~ +5	−20 ~ +10
Stages (Rectifier)	10	5	2	3	6
Maximum *PCE* (Rectifier)	42.8%@ *P_in_* = −16 dBm	—	78.2%@ *P_in_* = −12 dBm	73.9%@ *P_in_* = 4.34 dBm	43.6%@ *P_in_* = −7 dBm
Output voltage (V) (Rectifier)	2.32	—	1.0	3.5	1.725
Maximum *PCE* (RF harvesting IC)	—	32.8%@ *P_in_* = −2 dBm	—	—	40.56%@ *P_in_* = −6 dBm
Output voltage (V)(RF harvesting IC)	—	1.77	—	—	1.50
Load resistor (kΩ)	500	500	—	2	30
Chip area (mm^2^)	0.0296	16.56	—	0.954	0.2842
Power (μW)	—	—	—	—	42

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
