# Peer review of "Small-Area Radiofrequency-Energy-Harvesting Integrated Circuits for Powering Wireless Sensor Networks"

_sensors, 2019, doi:10.3390/s19081754_

Round 1

Reviewer 1 Report

The paper shows good details for designing a RF energy harvesting IC. The presentation is good. The novelty is not so significant.

There are few points to be highlighted as follows:

VA and VB can be varied due to PVT. How do the authors to tackle this?

Voltage reference, implemented using MOS weak inversion region, is subject to variation under process variation. The output voltage can be substantial in variation. How do the authors to address the precision issue?

For the frequency compensation in the LDO Regulator. The capacitor is CF but not Cz as stated in the manuscript. The authors need to inform the readers that whether it is output capacorless or output capacitor based design.

Reviewer 2 Report

This paper presents the design and implementation of an integrated circuit consisting of several building blocks - (1) RF-direct current (DC) rectifier, (2) an overvoltage protection circuit, (3) a low-power low-dropout (LDO) voltage regulator, and (4) a charger control circuit but not any sensor related building blocks. The authours have only mentioned " sensor in the title and abstract but not anywhere in the paper. This is a misleading issue. Despite this, I can recommend reconsideration if address the following changes/revision. 1. The authors must add an example of a wireless sensor in their paper. The sensor can be off-chip and powered up using the proposed energy harvesting system. 2. The characterization and experimental results of the connected sensor to the chip are demonstrated and discussed and its functionality is shown, 3. Authors should review all practical progress of energy harvesting for wireless sensors. In this review, the commercially available systems/devices are mentioned. A table should be added to compare the challenges addressed in literature and state of art technology.

Round 2

Reviewer 2 Report

This paper presents a RF energy harvesting IC suitable for sensors applications. Despite the fact that the focus of the paper, still is placed on the circuit design, I recommend the paper for publication after minor revision:

Add a table of IC facts including the energy consumption and energy delivery and other electrical characterization facts.

Add a short discussion section describing the advantage of this work for sensing application, and their challenges
